Methods

# Visualisation of *Euglena gracilis* organelles and cytoskeleton using expansion microscopy

Anežka Konupková ⓘ, Priscila Peña-Diaz ⓘ, Vladimír Hampl ⓘ

**This article explores the use of expansion microscopy, a technique that enhances resolution in fluorescence microscopy, on the autotrophic protist *Euglena gracilis*. A modified protocol was developed to preserve the cell structures during fixation. Using antibodies against key cytoskeletal and organelle markers, α-tubulin, β-ATPase, and Rubisco activase, the microtubular structures, mitochondria, and chloroplasts were visualised. The organisation of the cytoskeleton corresponded to the findings from electron microscopy while allowing for the visualisation of the flagellar pocket in its entirety and revealing previously unnoticed details. This study offered insights into the shape and development of mitochondria and chloroplasts under varying conditions, such as culture ages and light cycles. This work demonstrated that expansion microscopy is a robust tool for visualising cellular structures in *E. gracilis*, an organism whose internal structures cannot be stained using standard immunofluorescence because of its complex pellicle. This technique also serves as a complement to electron microscopy, facilitating tomographic reconstructions in a routine fashion.**

## Introduction

Expansion microscopy has emerged as a powerful technique for studying the morphology of cells and their organelles using fluorescence microscopy with increased resolution. It has been successfully applied to mammalian cells and tissues (Chen et al, 2015; Ku et al, 2016; Hümpfer et al, 2024) and to many single-cell eukaryotes across the eukaryotic tree including genera *Giardia*, *Plasmodium*, *Trypanosoma*, *Leishmania*, *Chlamydomonas*, *Trichomonas*, and *Eutreptiella*, and various dinoflagellates. In these microbial eukaryotes, this technique has proved useful in revealing the spatial organisation of their cytoskeleton, invasion apparatus, and the localisation of the translocons on the membrane of the hydrogenosomes, organelles related to mitochondria (Halpern et al, 2017; Gambarotto et al, 2019; Bertiaux et al, 2021; Gorilak et al, 2021; Makki et al, 2024; Mikus et al, 2024 *Preprint*). In this

report, we demonstrate the application of expansion microscopy to visualise the cytoskeleton, mitochondria, and chloroplasts of *Euglena gracilis*. *E. gracilis* is a free-living autotrophic protist with biotechnology potential (Ebenezer et al, 2022), and it is a model species of Euglenophyceae, a group of interest in the study of eukaryotic evolution and plastid endosymbiosis. The organisation of the euglenid cytoskeleton has been described through transmission and scanning electron microscopy (Piccinni & Mammi, 1978; Willey & Wibel, 1985; Surek & Melkonian, 1986; Farmer & Triemer, 1988; Shin et al, 2002). It starts in the region of basal bodies of the anterior and posterior flagellum. From these, three microtubular roots extend and initiate microtubules underlying the flagellar pocket and continue into microtubules of the pellicle. The pellicle is a complex proteinaceous structure underlying the plasma membrane and providing the cell with a firm but flexible surface capable of a specific form of movement known as metaboly (Cavalier-Smith, 2017; Leander et al, 2007; Esson & Leander, 2006; Sommer, 1965; Bricheux & Brugerolle, 1987; Dubreuil & Bouck, 1988). It is likely that the presence of this surface barrier prevents the staining of internal structures, apart from the pellicle and flagellum (Mermelstein et al, 1998; Nasir et al, 2018), with antibodies. This is evident from the absence of immunofluorescence studies in this organism. In the last 60 yr, the metabolism of *E. gracilis* has been thoroughly mapped (Schwartzbach & Shigeoka, 2017) and this knowledge is now supplemented using omics methods that provide genetic components (O'Neill et al, 2015; Yoshida et al, 2016; Ebenezer et al, 2019; Cordoba et al, 2021; Chen et al, 2024) and data on the protein composition of several basic compartments, mitochondria (Hammond et al, 2020), chloroplast (Novák Vanclová et al, 2020), and flagellum (Hammond et al, 2021). The metabolism of *E. gracilis* is dependent on growth conditions including the presence and type of organic carbon sources, oxygen, and light (Schwartzbach & Shigeoka, 2017), all of which clearly affect morphology and relative volumes of mitochondria and chloroplasts (Pellegrini, 1980a, 1980b). Unfortunately, the field is far from connecting these morphological changes to physiology also because it is difficult to assess them routinely in a larger collection of cells with a sufficient resolution. The main aim of this study was to validate expansion microscopy as a method for immunolabelling internal

---

Faculty of Science, Department of Parasitology, BIOCEV, Charles University, Vestec, Czech Republic

Correspondence: vlada@natur.cuni.cz

structures in *Euglena* and explore its usefulness for monitoring changes in the size and shape of chloroplasts and mitochondria.

## Results

To expand the cells of *E. gracilis*, we adopted a modified version of the protocol developed by Gorilak et al (2021). In this approach, the cells are fixed in solution before they come in contact with glass, preventing the deformation of cell shape that may occur when the cells adhere to the glass. We have tested various modifications of this protocol for *E. gracilis* cultures grown in different media. At first, we assessed three buffers for fixation, namely, PBS, 25 mM sucrose buffer, and the growth media itself (specified in the Materials and Methods section). The latter resulted in the best-preserved cell shape in all media tested—Cramer–Myers's media (CMM; Cramer & Myers, 1952), CMM with 0.4% (vol/vol) ethanol (CMMEt), and Hutner's media (HM; Hutner et al, 1966). Therefore, fixation in the growth media was used in all experiments. Unless stated otherwise, we used cells grown in CMMEt.

The expansion factor of the cells was determined by measuring the distance between the rows of pellicular microtubules in the central region of the cell on the confocal planes parallel to their course (N = 52; 0.851 [±0.098] $\mu m$) (Fig 1A and B, Table S1). By the single row of pellicular microtubules, we mean a group of four microtubules localised at the joint of neighbouring pellicular strips (Fig 1E; Cavalier-Smith, 2017; Bricheux & Brugerolle, 1987). These four microtubules appear in expansion microscopy images as a single line, and the distance between lines represents the width of pellicle strips. Each of the 52 measured cells was represented by an average of 20 measurements. The second size marker used to determine the expansion factor was the width of the axoneme in the straight central regions of the flagella (N = 52; 0.713 [±0.076] $\mu m$; Fig 1A and C, Table S2). Each of the 52 measured cells was represented by an average of 10 measurements. These values were compared with the dimensions measured on electron micrographs of unexpanded, chemically fixed cells (Fig 1E and F). On these micrographs, the distance between pellicular microtubules was measured as the distance between microtubule nr. 2 of adjacent pellicle strips. N represents the number of measured pairs. The electron micrographs revealed a distance of 0.270 $\mu m$ (±0.046 $\mu m$; N = 150) between pellicular microtubules and an axoneme width of 0.192 $\mu m$ (±0.007 $\mu m$; N = 10), resulting in cell expansion factors of 3.2 and 3.7x, respectively. The gel expansion factor was determined to be 4.5x (±0.1; N = 9; Table S3) based on photographs of gels before and after expansion. Expansion evenness was assessed by determining the aspect ratio of the major and minor axes of ellipses fitted into the peripheral microtubules of the proximal region of the anterior flagella. This region does not contain a central pair of microtubules, and it is wider in comparison with the axoneme of the protruding flagellum. Measurements taken from ten selected cross-sections revealed that the aspect ratio was 1.08 (±0.04; Fig 1D, Table S4).

### Visualisation of the microtubular cytoskeleton

Immunostaining of cells with an anti-$\alpha$-tubulin antibody revealed the prominent microtubular structures in the cell, namely, pellicular microtubules, flagellar microtubules, and the cytoskeleton lining the flagellar pocket (Fig 1A). In the detailed ventral-right view of the flagellar pocket (Fig 2A), confocal plane sections (Figs 2B and S1), and animation of the cell (Video 1), the principal components of *E. gracilis* cytoskeleton are visible. These are composed of two basal bodies (1, 2) bearing the posterior or ventral flagellum (PF) and anterior or dorsal flagellum (AF), respectively. The posterior flagellum is short and does not extend outside the flagellar pocket. Note the inflated transition zone (TZ), particularly in the anterior flagellum (Fig 2A, Video 1). Two microtubular roots are associated with basal body 1, the ventral root (VR), and the intermediate root (IR; Fig 2A and B section 1). The VR initially travels in the ventral-left direction and then loops around the bottom of the flagellar pocket directing right-anteriorly and flanking the flagellar pocket in the region free of microtubules. The distal part of the VR detaches from the flagellar pocket and stretches dorso-anteriorly towards the pellicle (Fig 2A and B section 8). The IR originates in the region between the two basal bodies, travels leftwards, and then continuously changes its course in a right-anterior direction around the flagellar pocket. A sheet of ventral pellicular microtubules originates in the proximity of the distal part of the IR and supports the membrane at the ventro-anterior side of the pocket (Fig 2A and B sections 9–10). The dorsal root (DR) originates at the dorsal side of the basal body 2 (Fig 2B section 1), extends leftwards, and for some distance travels closely along the IR (Fig 2A and B sections 3–7). Four bands of dorsal pellicular microtubules (Fig S1, arrows in sections 77, 89, and 107) separate from different parts of the DR, the first at the very proximal part close to the basal body and the others more distantly (Fig S1 sections 53 and 77). They all underlie the flagellar pocket from the dorsal side, and in the anterior region, they merge with one another forming a complete corset together with the ventral pellicular microtubules that lines the canal (Ca) connecting the flagellar pocket with the external environment (Fig 2A and B sections 9–12). At the end of the canal, these microtubules probably continue into pellicular microtubules (Pe). A spiral of para-reservoir microtubules reinforces the lining of the canal (Fig 2A and B section 11). Fig 2C–E displays mitotic cells in various stages. In these cells, the rows of pellicular microtubules are doubled to be distributed into daughter cells upon cytokinesis. In Fig 2C, the mitotic spindle (MS) is formed and the flagella are multiplied to four, all within a single flagellar pocket, of which only one, presumably the old anterior flagellum, reaches out of the canal. In Fig 2D and E, the flagellar pocket divides but remains connected in the canal region. The young anterior flagellum continuously grows through the canal to reach the outside of the cell. The mitotic spindle divides the chromosomes, which are not stained.

### Morphology and development of mitochondria

Immunostaining of *E. gracilis* with an anti-$\beta$-ATPase antibody enabled the visualisation of robustly stained mitochondria, and with lower intensity also chloroplasts because of the recognition of the plastidial ATPase by the same antibody (Table S6). In most cases, the signal from chloroplasts could be filtered out using deconvolution, which revealed primarily the morphology of mitochondria. We tested whether cells grown in the three most common media,

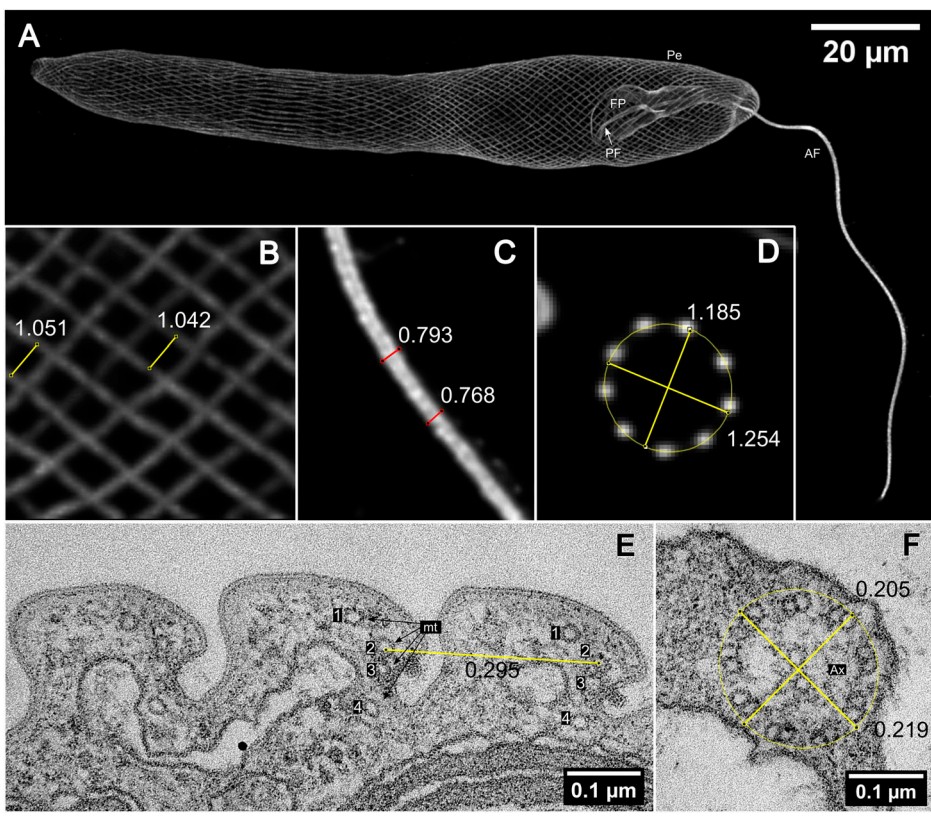

**Figure 1. Overview of the microtubular cytoskeleton of *E. gracilis* and measurements of expansion factor and evenness.**
**(A)** Expanded *E. gracilis* cell stained with the anti-α-tubulin antibody highlighting the major microtubular structures, pellicle (Pe), microtubules supporting the flagellar pocket (FP), the short posterior flagellum (PF), and the anterior flagellum (AF) extending outside the pocket. **(B)** Expansion factor was determined by measuring the distance between pellicular microtubules in the central part of the pellicle. **(C)** Second independent determination of the expansion factor was conducted by measuring the width of the axoneme in the straight region of the AF outside the pocket. **(D)** Evenness of expansion was assessed by measuring the aspect ratio between the major and minor axis of an ellipse fitted into the perpendicular section of microtubules in the transition zone of the anterior flagellum. **(E, F)** Electron micrographs of unexpanded *E. gracilis* pellicle structure (E) were used to measure the distance between microtubule (mt) nr. 2 and axoneme (Ax) width (F) allowing for the determination of the nonexpanded dimension of these structures after fixation for electron microscopy. All the illustrative measurements are expressed in *μ*m.

CMM, CMMEt, and HM, differ in mitochondrion morphology (Fig S2A). In all cases, the mitochondrion displayed the shape of a single reticulum (Video 2, Fig S2A). Mitochondria of photoheterotrophically grown cells in HM and CMMEt were mostly localised under the cell surface with a few threads extending towards the interior. The reticulum of phototrophically grown cells in CMM displayed less definition likely caused by the strong signal of the chloroplastic β-ATPase and was more evenly spread throughout the volume of the cell (Video 2). For at least 10 selected cells grown on each media, the relative volume of mitochondrion was calculated. These cells demonstrated significant variability in the mitochondrial volume fraction relative to the total cell volume: CMMEt from 6.4% to 16.9%, HM from 5.7% to 16.8%, and CM from 3.7% to 20.63% (Table S5). Note that different intensity thresholds were required to measure mitochondria in the cells grown in CMM, because of the stronger signal of their chloroplastic β-ATPase.

All following experiments were performed using asynchronous photoheterotrophic cultures maintained in CMMEt with a 12-h/12-h light/dark cycle. The appearance of the mitochondrial reticulum displayed variation as the culture aged (Fig 3) and with the light/dark cycle phase (Fig 4). The comparison of cultures of different ages was conducted through immunostaining of cells collected 3 h after beginning the light phase. Representative cells are displayed in Fig 3. On day 3, the mitochondrion formed a web of fine threads spreading relatively evenly under the whole cell surface. By days 5 and 7, the threads became thicker, displaying wide but flat connections, with lobes in some areas. However, there were relatively large areas beneath the cell surface that lacked mitochondria. By

day 10, threads became finer again but not to the degree observed on day 3, and there were fewer in number. The development of mitochondria was observed over a single day on the 7-d-old culture, which was followed from time 0 h (end of the dark phase) to time 12 h (end of the light phase). Representative cells are shown in Fig 4. It is important to note that cell division in this culture is not synchronised with the light/dark cycle; therefore, the changes observed should not be influenced by the cell cycle phase. At the end of the dark phase (0 h), the mitochondrial reticulum consisted of fine threads, though neither as fine nor numerous as those seen on day 3. As the hours progressed (from 3 to 12 h), the threads became compact, forming dense, interconnected regions with wide, flat lobes.

## Morphology and development of chloroplasts

Chloroplasts were observed using immunostaining with an anti-Rubisco activase antibody (anti-RCA; Table S7), which strongly stained the pyrenoids. In most cells examined, the remaining volume of the chloroplasts displayed a weaker but well-defined signal. The antibody also produced unspecific signals on the pellicle and around some cells. Chlorophyll autofluorescence was not detected (Fig S2C). The morphology of chloroplasts was visibly affected by the culture media (Fig S2B). Chloroplasts in cells grown phototrophically on CMM (without any carbon source) were larger, with intensely stained pyrenoids and more jagged edges than those in photoheterotrophically grown cells on CMMEt and HM. For six selected cells from each medium, the relative volume of

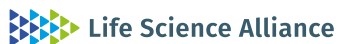

**Figure 2. Closer look at the microtubular ultrastructure of the flagellar pocket in *E. gracilis* and at cells in various stages of mitosis.**
**(A, B)** Overview of the flagellar pocket of *E. gracilis* stained with the anti-α-tubulin antibody (A) accompanied by selected Z-stack slices from the transverse section through the anterior part of the cell (B1–12). The complete Z-stack can be obtained from https://doi.org/10.6084/m9.figshare.27198924.v1; the sections correspond to the following slices of the Z-stack: 11, 22, 32, 40, 52, 63, 83, 102, 117, 120, 126, and 133. Two basal bodies (1, 2) bear anterior (AF) and posterior flagellum (PF), both featuring inflated transition zones (TZ). These basal bodies are associated with the ventral root (VR), dorsal root (DR), and interior root (IR). The ventral pellicular microtubules (VPM) and dorsal pellicular microtubules (DPM) underlie the flagellar pocket and the canal (Ca), which is also surrounded by para-reservoir microtubules (PMT). Outside of the canal, VPM and DPM continue into the pellicular microtubules (Pe). **(C, D, E)** *E. gracilis* cells in various stages of mitosis, revealing the mitotic spindle (MS) and the doubling of the pellicular microtubules, along with the duplicated anterior and posterior flagella in images (D, E).

chloroplasts was measured. The relative volume of chloroplasts in cells grown in CMM varied from 7.6% to 13.7% of the total cell volume and was visibly larger than in cells grown in HM (4.5–8.5%) and CMMEt (5.3–8.7%; Table S5). It is important to note that the intensity threshold used for measuring the relative volume of chloroplasts in CMM-grown cells was higher in most cases because of the stronger signal of the anti-RCA antibody in these cells.

We monitored changes in the morphology of chloroplasts at various ages of culture (Fig 5) and throughout the light cycle of

1 d (Fig S3). This was also performed using asynchronous photo-heterotrophic cultures maintained in CMMEt with a 12-h/12-h light/dark cycle. Morphologies of chloroplasts in cells from 3- to 10-d-old cultures were observed on cells sampled 3 h after the beginning of the light phase (Fig 5). On day 3, the chloroplast signals were barely visible, indicating low abundance of Rubisco activase. From day 5, the shape of chloroplasts was clearly discernible, and from day 7, the pyrenoids in the centre of each chloroplast displayed intense staining. We also observed chloroplast development throughout a

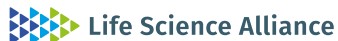

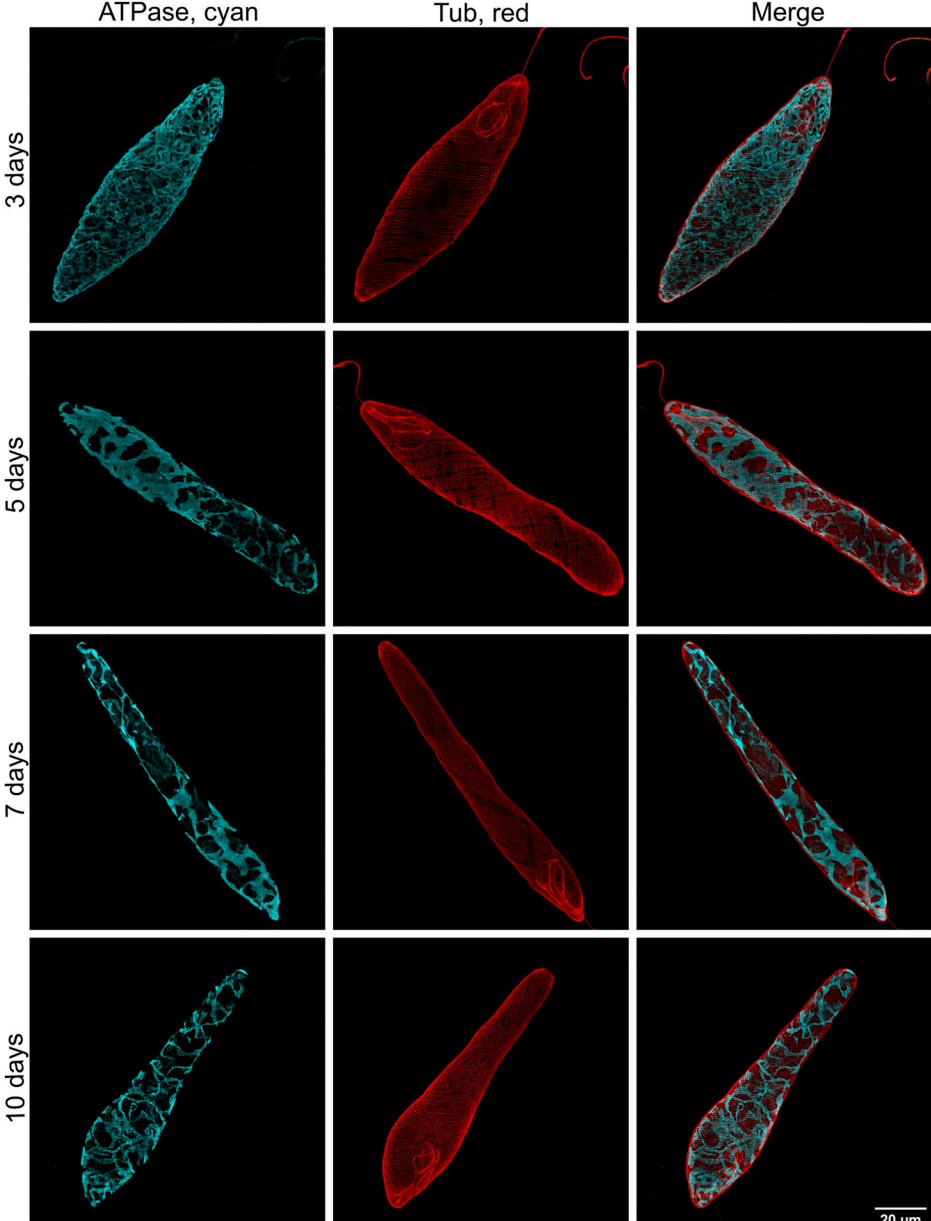

**Figure 3. Mitochondrial morphologies in representative cells of *E. gracilis* collected at different ages of the culture.**
Mitochondria in *E. gracilis*, stained with the anti-β-ATPase antibody, visualised in cyan, display morphological changes across cultures that are 3-, 5-, 7-, and 10-d-old, grown in CMMEt. The red channel represents the cell shape, outlined by the pellicle stained with the anti-α-tubulin antibody. The merged channels highlight the subpellicular localisation of the mitochondria in *E. gracilis* cells.

single day using a 7-d-old culture, tracking the changes from time 0 h (end of the dark phase) to time 12 h (end of the light phase). Representative cells are shown in Fig S3. The changes in chloroplast morphology were not as pronounced as those in mitochondria. Although chloroplasts and pyrenoids were visible at all time points, the signals appeared to be strongest during the middle of the day at 3, 6, and 9 h.

## Discussion

Our study demonstrates that the method of expansion microscopy, which has been successfully applied to many types of cells, is also suitable for visualising the cytoskeleton and organelles of autotrophic protist *E. gracilis*. In addition, it allows for quantitative measurements of the volumes of these organelles. We adapted the protocol of Gorilak et al (2021) and successfully immunostained microtubules with anti-α-tubulin, mitochondria with anti-β-ATPase, and chloroplasts with anti-Rubisco activase antibodies. The anti-β-ATPase antibody stained β-ATPase in both mitochondria and chloroplasts; however, the chloroplast signal was generally weaker and easily filtered out by deconvolution. The specificity of the anti-Rubisco activase antibody was not absolute, it stained pyrenoids more intensely than other parts of chloroplasts, and produced some unspecific signal in the cytoplasm and outside of the cell. Nonetheless, in both cases the specific signal was strong, making these antibodies useful markers for mitochondria and chloroplasts.

| ATPase, cyan | Tub, red | Merge |
|---|---|---|

**0h**

**3h**

**6h**

**9h**

**12h**

20 μm

**Figure 4. Mitochondrial morphologies in representative cells of *E. gracilis* collected at different time points during the light phase.** The cultures were maintained under 12 h of light followed by 12 h of darkness. Cells were fixed at 0, 3, 6, 9, and 12 h after the beginning of the light phase. The network of the mitochondria, stained with the anti-β-ATPase antibody (shown in cyan), displays morphological changes during these time points. The red channel depicts the pellicle stained with the anti-α-tubulin antibody.

Notably, the chlorophyll autofluorescence was not observed, which is consistent with reports on other algae (Mikus et al, 2024 *Preprint*).

The determined cell expansion factor (3.2-3.7x) was lower than the values reported for kinetoplastids (4.7-6.0x) using a similar protocol. In contrast, the expansion factor of the gel (4.5x) was almost identical to the previously reported value (4.7x) (Gorilak et al, 2021). This difference may be caused by the resistance of *E. gracilis* cells to hydrolysis and expansion, attributed to the presence of a firm pellicle. Optimising the process in this direction could

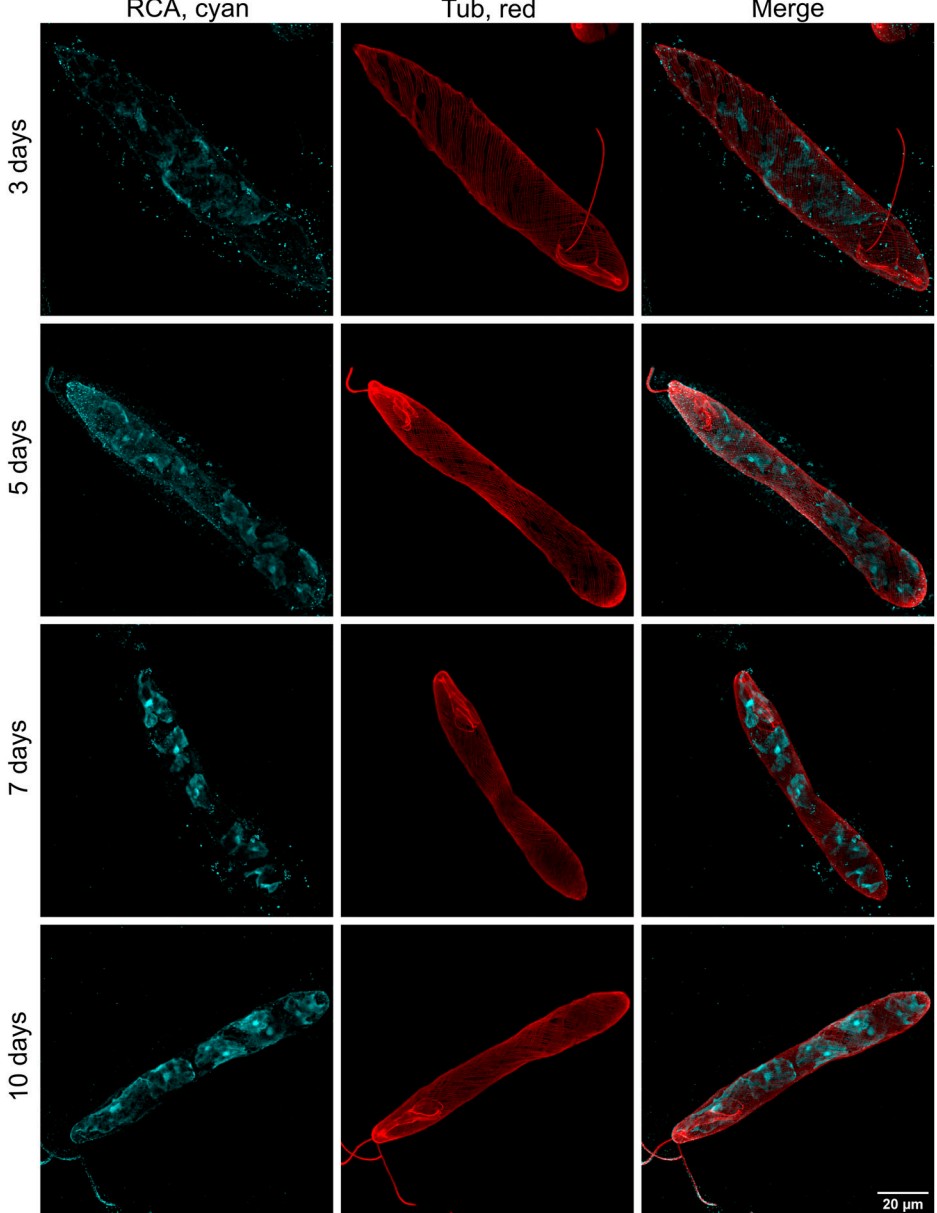

|  | RCA, cyan | Tub, red | Merge |
|---|---|---|---|
| 3 days | | | |
| 5 days | | | |
| 7 days | | | |
| 10 days | | | |

20 μm

**Figure 5. Chloroplast morphologies in representative cells of *E. gracilis* collected at different ages of the culture.**
The expanded chloroplasts of *E. gracilis* were stained with an anti-RCA antibody (cyan), which accumulated primarily in the pyrenoids. Morphological changes in chloroplasts were observed between 3-, 5-, 7-, and 10-d-old cultures grown in CMMEt. The signal of chloroplasts appears to be the lowest and the unspecific signal the highest in a 3-d-old culture, whereas older cultures display a stronger chloroplast signal, particularly in pyrenoids. The red channel indicates the cell shape, outlined by the pellicle stained with the anti-α-tubulin antibody.

be beneficial; however, the current protocol already provides sufficient resolution for many purposes, thanks to the large size of *E. gracilis* cells.

Staining with an anti-α-tubulin antibody enabled visualisation of the flagellar apparatus and associated microtubular structures. The spatial organisation of these structures (Figs 2 and S1) corresponds to reports based on electron microscopy (Piccinni & Mammi, 1978; Willey & Wibel, 1985; Surek & Melkonian, 1986; Farmer & Triemer, 1988; Shin et al, 2002). The cell contains two flagella of which the anterior flagellum (AF) protrudes outside the flagellar pocket and is used for movement, whereas the posterior flagellum terminates inside the pocket. The widths of transition zones, regions composed of nine microtubular doublets without a central pair of microtubules, along with the distal parts of the basal bodies and proximal

regions of the axoneme in both flagella, were visibly inflated. This inflation is especially noticeable in the AF, where it reaches up to one-third of the length hidden in the flagellar pocket (Fig 2A and B). In this region, the peripheral doublets form a circle ~1.5x wider than the axoneme in the distal part of the flagellum. It is known that the transition zones of Euglenida are wider, hollow, and relatively long (Piccinni & Mammi, 1978; Kivic & Walne, 1984; Farmer & Triemer, 1988; Simpson, 1997). The absence of internal structural components in their interior likely contributes to a decrease in structural integrity, resulting in a disproportionate broadening during expansion. Cross-sections of this area were used to estimate the evenness of expansion (Fig 1D). Expansion microscopy allowed us to visualise the course of three microtubular roots, VR and IR associated with the basal body of the PF, and DR associated with the

basal body of the AF. The relative spatial organisation and trajectory through the cell correspond to observations from electron microscopy studies (Surek & Melkonian, 1986). A significant advantage of expansion microscopy is its ability to provide confocal plane sections (Figs 2B and S1), which are analogous to serial electron micrographs with smaller resolution. It also greatly facilitates the creation of 3D reconstructions (Fig 2A, Video 1). This comprehensive view of the structures is invaluable as it helps to discern features not easily recognisable by EM. One example is the split of the proximal region of dorsal pellicular microtubules in four distinct bands separating from different parts of DR (Fig 2A and B section 5–7, Fig S1 sections 53, 77, 89, and 107). To our knowledge, this feature has never been previously described.

The introduction of antibodies targeting organellar markers, the anti-$\beta$-ATPase antibody for visualising mitochondria and the anti-Rubisco activase antibody for chloroplasts, enabled detailed observation of the morphology of these organelles. It allowed also for the quantification of their relative volumes within cells and monitoring changes throughout the light cycle and during the culture's growth. Because the cultures were asynchronous, the observations should not have been influenced by the cell cycle. As shown previously, the mitochondrion exhibited the shape of a single reticulum. The configuration of this reticulum was influenced by the cultivation media (CMM, CMMEt, and HM). Cells grown photoautotrophically (CMM) displayed a reticulum that spanned the entire cell volume, consistent with findings from electron micrographs (Pellegrini, 1980a). In contrast, the mitochondria of photoheterotrophically grown cells (CMMEt and HM) were primarily located just beneath the cell surface. Unlike Pellegrini's earlier observations (Pellegrini, 1980b), these mitochondria did not extend many threads into the central region of the cells. This discrepancy might result from differences in the carbon source (sodium acetate) or temperature regime (cold/warm cycles) used by Pellegrini. The variability of the mitochondrial reticulum was also noted with respect to the culture age and light cycle, adding yet another factor to consider. The shape of the mitochondria varied from a network of numerous fine threads, observed particularly in the 3-d-old culture and at the end of the dark phase, to a network characterised by small, flat areas connected by fewer and often thicker connections. The presence of fine-threaded mitochondria correlated with a less intense chloroplast signal, as stained with the anti-Rubisco activase antibody, suggested that carbon fixation was occurring at lower levels during these time points. We anticipate that the changes in mitochondrial shape have physiological implications and may reflect changes in its biochemical performance; however, investigating the biochemical or molecular basis of this phenomenon is beyond the scope of this study. We measured relative volumes of mitochondrion and chloroplasts in six typical 7-d-old cells, 3 h after the beginning of the light cycle, grown on CMMEt, HM, and CMM. The relative volumes of mitochondria were surprisingly variable (6.4–16.9%; 5.7–16.8%; 3.7–20.63%), whereas chloroplast volumes were relatively stable (5.3–8.7%; 4.5–8.5%; 7.6–13.7%). We cannot rule out the possibility that the variation in volumes of mitochondria may have been due to a methodological artefact. Yet, consistent volumes of chloroplasts suggest that either the expansion factor of mitochondria varies markedly more than that of chloroplast, or the cells in the population exhibit considerable variability in mitochondrial volumes. We favour the latter, considering that the population was asynchronous and included cells at different stages of their life cycle. Estimates based on EM tomography obtained by Pellegrini (1980a, 1980b) indicate that under photoautotrophic conditions, that is, grown on inorganic salt CMM, mitochondria comprise roughly 6% of the cell volume and are independent of the cell cycle phase. After adding sodium acetate, the volume of mitochondria in photoheterotrophically grown cells increased to 10–11%. In cells grown heterotrophically in darkness, mitochondrial volume increased to 15–16%. The cells measured by us encompassed the entire range of values previously reported. Pellegrini (1980a) found that the relative volume of chloroplasts varied, with photoautotrophically grown cells in CMM showing a volume of 16.3%. In contrast, photoheterotrophically grown cells in CMM supplemented with sodium acetate exhibited a relative chloroplast volume of 12–13% (Pellegrini, 1980b). For cells grown heterotrophically in CMM with sodium acetate in the dark, the volume of chloroplasts dropped to 6.8% or less. Our measurements indicated that the volume of chloroplasts in cells grown on all media fell between the values observed for light- and dark-grown heterotrophic cells. It is not surprising that the appearance of chloroplasts varied with the cultivation media, culture age, and light cycle. Chloroplasts from cells cultivated photoautotrophically in mineral CMM were larger than those from cells grown in both CMMEt and HM. In addition, in the CMMEt-grown cells, the signal of Rubisco activase was generally weakest in young cultures and at the beginning of the light phase, which likely corresponds to the lowest levels of photosynthetic activity.

Expansion microscopy has proven to be an effective method for visualising the cytoskeleton and organelles in *E. gracilis*. Currently, it is the only functional alternative to immunofluorescence for this organism, as there is no other functional protocol for immunofluorescence staining of structures within the cell. This limitation is likely due to the presence of a compact pellicle that prevents antibody penetration. However, hydrolysis and subsequent expansion solve this problem, enabling immunofluorescence staining of mitochondria, plastids, and microtubule components associated with the flagellar pocket. Moreover, expansion microscopy provides a useful means of obtaining 3D tomographic reconstructions of stained structures. Regardless that its resolution falls short of that of electron microscopy, it still allows for clear visualisation of cytoskeletal and other cellular structures. Consequently, this method serves as a valuable complement to electron microscopy.

# Materials and Methods

### Cell culture

*E. gracilis* strain Z was cultured photoautotrophically in the Cramer–Myers's media (CMM; Cramer & Myers, 1952), pH 6.8, or photoheterotrophically in CMM with the addition of the 0.4% (vol/vol) ethanol as a carbon source (CMMEt), or in Hutner's media (HM; Hutner et al, 1966), pH 3.5. The cultures were maintained in 25-$cm^2$ culture flasks with vented caps containing 10 ml of media and kept

in a Q-Cell 60 incubator (Pol-Lab) at 25°C, with a light and dark cycle of 12 h each.

### Co-immunoprecipitation

Co-immunoprecipitation was performed to assess the specificity of the primary antibodies used for immunostaining. For this purpose, *E. gracilis* cultures grown in CMMEt medium for ~5–7 d were used. 1 × 10⁷ cells were centrifuged at 1,000*g* for 10 min at 4°C to eliminate the media, and the pellet was resuspended in a maximum of 3 ml of 50 mM Tris–HCl, pH 8, 150 mM NaCl, and protease inhibitors (Roche). The cell suspension was then lysed by sonication, using a Qsonica sonicator Q125 with a 1/8″ probe (Newtown, CT, USA) at five pulses on, two pulses off, for 2 min at 40% amplitude. Breakage of cells was verified microscopically. The cell lysate was centrifuged at 14,000*g* for 10 min at 4°C, and the supernatant was used with an Immunoprecipitation Kit—Dynabeads Protein G (Invitrogen)—according to the manufacturer's instructions. The protocol was followed until the final beads were washed, which was followed by three more washes with PBS, to finally be directly stored at –20°C for subsequent proteomic analysis performed by the core facility.

### Expansion microscopy

#### *Sample preparation*
The cells were harvested 3 h after the beginning of the light phase (unless otherwise specified) and counted using Beckman Coulter Z2 Cell and Particle Counter (Beckman Coulter, Inc.). A total of 1 × 10⁶ cells were pelleted by centrifugation at 400*g* for 30 s and gently resuspended in 500 µl of fixative solution. Three types of fixative solutions were tested, consisting of 4% (vol/vol) formaldehyde and 4% (wt/vol) acrylamide dissolved either in PBS as used in Gorilak et al (2021), or in 25 mM sucrose solution (25 mM sucrose; 10 mM Hepes-KOH, pH 7.5; 2 mM MgSO₄) as used in Heiss et al (2024 *Preprint*) or in three different cultivation media—CMM, CMMEt, and HM. The overall cell shape, flagellum, and pellicle strips were best preserved in preparations fixed in cultivation media, which was therefore used for all subsequent preparations (data not shown). The cell suspension was pipetted onto a clean, round coverslip coated with poly-L-lysine in a 24-well plate and left to settle down in the dark at RT overnight.

The gel preparation was performed according to Gorilak et al (2021). After overnight incubation, the coverslip was gently washed with PBS and quickly transferred onto a 50 µl drop of monomer solution. This solution consisted of 19% (wt/vol) sodium acrylate, 10% (wt/vol) acrylamide, and 0.1% (wt/vol) N, N′-methylenebisacrylamide in PBS, with the final addition of cold 0.5% (wt/vol) ammonium persulphate and 0.5% (vol/vol) N, N, N′, N′-tetramethylethylenediamine (for gelation) on parafilm placed in a wet chamber on ice for 5 min, and then, the whole chamber was transferred to 37°C for 30 min. After gelation was complete, the coverslip with the gel was removed from the parafilm and transferred into 1 ml of denaturation buffer (50 mM Tris–HCl, pH 9.0, 200 mM NaCl, 200 mM SDS) in a 12-well plate. The plate was left at RT for ~10 min with gentle shaking until the coverslip detached from the gel. The gel with the remaining denaturation buffer was transferred to a 1.5-ml tube containing 0.5 ml denaturation buffer and boiled for 1 h at 95°C. After denaturation, the

gel was washed three times with Milli-Q water in a ∅ 10-cm Petri dish, resulting in the first gel expansion.

The expanded gel was cut into small squares of ~1 × 1 cm and transferred to a 24-well plate. The gel was blocked with PBS, 2% (wt/vol) BSA for 20 min at RT. After blocking, primary antibodies were diluted in appropriate concentrations in 300 µl PBS, 2% (wt/vol) BSA and added to the well with the gel. The plate was incubated overnight with gentle shaking in 4°C. The used antibodies were anti-α-tubulin, raised in guinea pig (anti-αTub; AA345; ABCD Antibodies) at a dilution 1:300; anti-β-ATP synthase of *Trypanosoma brucei*, raised in rabbit (anti-β-ATPase; kindly provided by Alena Zíková) at a dilution 1:300; and anti-Rubisco activase, made in rabbit (anti-RCA; AS10 700; Agrisera) at a dilution 1:300.

The next day, the liquid from the plate was discarded and the gel was washed 3x with PBS for 20 min at RT. A solution containing secondary antibodies was prepared, using 300 µl per gel in PBS, 2% (wt/vol) BSA, at a dilution 1:300. This solution was incubated overnight at 4°C with mild shaking, and the mixture was covered with aluminium foil. The secondary antibodies used were Alexa Fluor 488, goat anti-rabbit IgG, A-11008; Alexa Fluor 488, goat anti-guinea pig IgG, A-11073; Alexa Fluor 594, goat anti-guinea pig IgG, A-11076. All secondary antibodies were purchased from Invitrogen. Finally, the gels were washed twice for 20 min with 1 ml of PBS each and then expanded by 3 × 20 min washes in 1 ml of Milli-Q water.

#### *Confocal microscopy*
Before microscopy, the gel was transferred onto a poly-L-lysine–coated 35-mm glass-bottom dish. Imaging was performed using a Nikon CSU-W1 (Nikon Instruments Inc.) spinning disc microscope with a CF Plan Apo VC 60XC WI objective (with water immersion). The NIS-Elements 5.42 software was used for capturing images. For excitation, laser wavelengths of 488 and 561 nm were employed for Alexa Fluor 488 and 594, respectively, using a single quad-band dichroic mirror (405/488/561/638 nm) to avoid channel shift. The excitation parameters, including laser intensity and exposure time, were adjusted before each capture to achieve optimal contrast without overexposure. Generally, excitation using intensity ~10–20 was performed for 50–200 ms (depending on the relay and tube lens magnification—1x, 1.5x, 2x, or 3x) for the 488-nm excitation laser. For the 561-nm excitation laser, intensity 40–50 was used for 100–400 ms. The pinhole size was set to 50 µm. Image capture was performed with a PRIME BSI camera (Teledyne Photometrics) with 2048 × 2048 pixels and pixel size 6.5 × 6.5 µm. The Z-stack step size ranged from 0.1 to 0.3 µm, and pixel size varied according to the magnification, 112 × 112 nm, 74.7 × 74.7 nm, 56 × 56 nm, and 37 × 37 nm for combined magnification of the relay and tube lens 1x, 1.5x, 2x, and 3x, respectively.

### Transmission electron microscopy

For ultrastructural analysis by transmission electron microscopy (TEM), 5-d-old cells grown on CMMEt were harvested and pelleted by centrifugation at 400*g* for 30 s. Cells were first fixed in a solution of 2.5% glutaraldehyde and 4% formaldehyde in 0.1 M cacodylate buffer for 1 h, then washed and embedded in 2% (wt/vol) low-melting agarose as a cell pellet. Dissected pieces (1 × 1 × 1 mm) of agarose-embedded pellet were then post-fixed in osmium tetroxide (1% in distilled water) for 1 h on ice, then in reduced osmium

(1% osmium tetroxide and 1.5% ferrocyanide in distilled water) for 1 h on ice. After washing with distilled water, samples were dehydrated in 50% and 75% ethanol (10 min each), then subjected to en bloc staining in 0.5% (wt/vol) uranyl acetate in 75% ethanol overnight at 4°C followed by three washes in 75% ethanol. The samples were further dehydrated in 85%, 95%, and 100% ethanol (10 min each) and subsequently in propylene oxide (two times for 20 min). Samples were then embedded in SPURR Low Viscosity Embedding Kit (EMS#14300) and cured at 70°C for 12 h. The ultrathin 60-nm sections were cut with a diamond knife using Leica EM UC7 ultramicrotome, collected on Cu slot grids with 1% formvar and 2-nm carbon, and post-contrasted with 3% lead citrate for 2 min. Imaging data were acquired on a JEM 2100 Plus transmission electron microscope (JEOL) operated at 200 kV using a TVIPS XF-416 camera.

### Data analysis

Images in Z-stacks were deconvolved using SVI Huygens Professional 22.10 software (Scientific Volume Imaging, http://svi.nl), except for the chloroplasts' visualisation, which was processed without deconvolution. Maximal projection of the whole Z-stack (or selected parts of the Z-stack for the visualisation of the flagellar pocket, cell division, or basal body cross-section) was created using Fiji open-source software (Schindelin et al, 2012). This software was also employed for further analysis of the transition zone's isotropic expansion and for calculating the expansion factors of the pellicle and axoneme. The 3D models were prepared using Imaris 9.9.1 and merged in Zoner Photo Studio X (Zoner a. s.) in Video 2.

The relative volumes of organelles were measured using NIS-Elements software (version 6.0.2) with AI functionalities. To measure the total cell volume, a binary layer was defined based on the signal from the anti-$\alpha$-tubulin antibody. In the binary editor, the binary layer was connected into a single object using the "Close" transformation, which combines dilatation (to enlarge the object) followed by erosion (to subtract the marginal pixels of the object). The signal from the flagellum was excluded from these measurements. For measuring the volumes of mitochondria and chloroplasts, the threshold 500 was set to define a binary layer using the signal of anti-$\beta$-ATPase and anti-RCA antibodies. In the case of cells grown in CMM, the threshold was adjusted to focus primarily on the signal from the organelles being measured (Table S5). These binary layers were connected into 3D objects, and their volumes were measured. Volumes below 1 $\mu m^3$ were considered background.

The supplementary material includes Figs S1, S2, and S3, Tables S1, S2, S3, S4, S5, S6, and S7, Video 1 and Video 2. Fig S1 presents selected transverse confocal plane sections of the anterior part of the cell containing the flagellar pocket. Fig S2A–C illustrates the morphologies of mitochondria and chloroplasts in representative cells grown in various growth media, along with negative controls for those cells. Fig S3 shows the development of chloroplasts at different time points (0, 3, 6, 9, 12 h) during the light phase of growth. Tables S1, S2, S3, S4, and S5 contain the raw data from the measurements used in the study, including graphs of pellicle microtubule distances and axoneme widths, which were used for calculating expansion factors. It also includes data for calculating the gel expansion factor, aspect ratios of ellipses fitted into the

perpendicular sections of microtubules in the transition zone of the anterior flagellum (used for the determination of the evenness of expansion), and the relative volumes of mitochondria and chloroplasts measured in individual cells. Tables S6 and S7 contain Western blot analysis and the proteomic confirmation of antibodies used in the study. Video 1 provides a detailed overview of the microtubular cytoskeleton of the flagellar pocket. Video 2 provides an overview of the network of expanded mitochondrion in various growth media stained with the anti-$\beta$-ATPase antibody.

## Data Availability

The data supporting all figures and measurements can be found in the published article, in its online supplemental material, and on Figshare.com.

## Supplementary Information

## Acknowledgements

This work was supported by the Czech Science Foundation project number 23-07277S to V. Hampl and the Grant Agency of Charles University project number 184924 to A. Konupková. The authors would like to acknowledge the Imaging Methods Core Facility at BIOCEV, which is supported by the Ministry of Education, Youth and Sports of the Czech Republic (document number LM2023050 Czech-BioImaging), for their assistance in this study. Special thanks to David Liebl for acquiring TEM data and to Zuzana Čočková for her help with data analysis. We also express our gratitude to Yi-Kai Fang for his initial assistance with expansion microscopy.

### Author Contributions

A Konupková: conceptualisation, data curation, formal analysis, investigation, visualisation, methodology, and writing—review and editing.
P Peña-Diaz: conceptualisation, supervision, and writing—review and editing.
V Hampl: conceptualisation, supervision, funding acquisition, and writing—original draft, review, and editing.

### Conflict of Interest Statement

The authors declare that they have no conflict of interest.

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
