## [Reviewer comments · Life Science Alliance]

Life Science Alliance

Visualization of *Euglena gracilis* Organelles and Cytoskeleton Using Expansion Microscopy

Anežka Konupková, Priscila Peña-Díaz, and Vladimír Hampl

DOI: <https://doi.org/10.26508/lsa.202403110>

Corresponding author(s): Vladimír Hampl, Charles University and Anežka Konupková, Charles University

Review Timeline:

Submission Date:	2024-10-23
Editorial Decision:	2024-12-02
Revision Received:	2025-01-26
Editorial Decision:	2025-01-27
Revision Received:	2025-01-28
Accepted:	2025-01-29

Transaction Report:

December 2, 2024

Re: Life Science Alliance manuscript #LSA-2024-03110-T

Dr. Vladimír Hampl
Charles University
Parasitology Department
Albertov 6
Prague 2 12800
Czech Republic

Dear Dr. Hampl,

Thank you for submitting your manuscript entitled "Visualization of *Euglena gracilis* Organelles and Cytoskeleton Using Expansion Microscopy" to Life Science Alliance. The manuscript was assessed by expert reviewers, whose comments are appended to this letter. We invite you to submit a revised manuscript addressing the Reviewer comments.

Thank you for this interesting contribution to Life Science Alliance. We are looking forward to receiving your revised manuscript.

Sincerely,

B. MANUSCRIPT ORGANIZATION AND FORMATTING:

Reviewer #1 (Comments to the Authors (Required)):

The study successfully applied expansion microscopy as a technique to visualize immunofluorescently labelled structures in *E. gracilis* at a level of detail and resolution that is comparable to transmission electron microscopy. The authors modified an existing expansion microscopy protocol to better preserve cell shape and ultrastructure. They labelled the cytoskeleton, mitochondria and chloroplasts of cultivated *E. gracilis* cells and described traits structures, such as the flagellar root system, that were consistent with previous electron microscopy studies of *E. gracilis*. This paper illustrates that it is feasible to use expansion microscopy as a method to visualize different cytoskeletal elements in cultivated euglenids, possibly in substitution for TEM, which is exciting.

However, the manuscript was not thoroughly edited prior to submission; it lacked clarity throughout and contained several spelling errors and grammatical mistakes (e.g., sentence structure, paragraph structure). It might be helpful for the authors to work with a professional editing service to revise the text and incorporate adequate scientific background for the results and discussion.

As described below, I am suspicious of some of the claims made by the authors, in part due to an obvious lack of relevant literature cited and a rationale for the specific techniques/experiments performed (e.g., comparisons of culture media, light regimes etc.). While these authors present beautiful and confirmatory micrographs of the cytoskeleton and organelles of *E. gracilis*, the manuscript is limited in terms of novel results.

My specific comments/suggestions are provided below:

Introduction

* The introduction of this manuscript is missing a significant amount of relevant information that would greatly improve the clarity of the paper. For example, the authors highlight that using expansion microscopy on *E. gracilis*, an organism that has been extremely well-characterized, could be useful, but they do not indicate any specific findings/ultrastructural traits/questions etc. that can be elucidated using expansion microscopy. From the introduction, the aim of this paper is unclear. It appears that these authors are trying to present a new protocol for expansion microscopy that can be successfully applied to visualizing the cytoskeleton, mitochondria and plastids of *E. gracilis*. There is no mention of testing various media/fixation buffers, or observing changes in organelle morphology during different phases of a light/dark cycle. I suggest that these authors flesh out the introduction with more background information to provide the necessary context for the rest of the paper and to highlight the aims of this study. For example, why might light/dark cycles influence mitochondrial ultrastructure? Why test this? This type of information should be made clear in the introduction of this manuscript.

* The literature cited in the introduction of the paper is limited, where several important statements are missing any citations, and some statements are missing key citations for relevant papers. For example:

"This technique has proved useful in visualising the spatial organisation of the cytoskeleton, invasion apparatus, and the localisation of the translocons on the membrane of the hydrogenosomes (mitochondrion-related organelles)."

Including citations for each of these three examples and providing information on what organisms were used for each example (i.e., the invasion apparatus of which organism, hydrogenosomes of which organism).

"The presence of this surface barrier, however, complicates the staining of internal structures with antibodies, making fluorescence microscopy of little use for this organism."

Provide citations for this statement. I would also suggest providing some information on how expansion microscopy mitigates this problem.

"The organisation of euglenid cytoskeleton..." and "The pellicle is a complex..."

Both of these sentences are missing several important citations that specifically address the unity, diversity and evolutionary history of traits associated with the pellicle.

Results & Methods:

- * The first two paragraphs of the results section read as a methods section.
- * The figures presented in this manuscript are of high quality and highlight the cytoskeletal and organellar complexity of *E. gracilis*. Many of the labels used in these figures are not cited in the text, and as a result, it was challenging to follow the text at times. Consider including a reference to the labels in the figures consistently throughout the text of the Results section.
- * The authors attempted to quantify ultrastructural differences in mitochondria and plastids as well as changes in carbon fixation rates by plastids, both in response to light limitation by fluorescently labelling these structures and imaging them via expansion microscopy.

It is strange that the authors did not observe any autofluorescence signal from the plastids of the cells given that *E. gracilis* cells are well known to autofluoresce, and no clearing step was indicated in the methods (as this would be necessary to clear the fixed cells of autofluorescence). Was a negative control used to determine whether or not there was autofluorescence present? Because chlorophyll autofluorescence would be observed using the excitation wavelengths that these authors used (488 nm), a negative control would be necessary to determine if there is/isn't autofluorescence. Including negative controls when performing fluorescence microscopy experiments is a standard for these types of experiments to substantiate the claim that autofluorescence was not present in the tissue/cells imaged.

* The authors specified that they used AlexaFluor 488 dye for each secondary antibody, meaning that the cytoskeleton, mitochondria and plastids of the prepared cells would all be labelled with the same fluorophore and would fluoresce under the same excitation spectra. If this is the case, describe how it was possible to obtain micrographs of the plastids, mitochondria and microtubular structures separately for the same cells - as it seems as though all three structures would fluoresce at once. Could this be an issue in how these methods were presented?

* The authors test several different media for fixation to select one that results in optimal ultrastructural preservation of cells; however, they do not show any micrographs of these different tests as part of the results. If the authors wish to conclude that one type of media is better at preserving cell morphology than others, it would be good to include micrographs that show the differences in cell preservation using the different types of media as supplementary material.

* The language used in the text is not always consistent with the literature. For example, using terms such as "turn/turning" or "dorso-anteriorly" when describing the flagellar root system is not consistent with the language used in other ultrastructural studies of euglenids. The authors also interchange the terms "immunolabeled" with "immunodecoration" and "immunostain". It would be clearer to maintain consistency with the terms used throughout the text.

* The text describing Figure 2C is limited to only one sentence. Consider describing these results in more detail.

Discussion

* The discussion of this manuscript either does not mention several findings that were highlighted in the results or only superficially attempts to place these results into the context of existing literature. For example, the localization of the mitochondria to the cell surface (in cells that are grown, I assume, aerobically) is unusual and, as far as I know, inconsistent with the previous EM characterizations of aerobic *E. gracilis*.

* These authors used an anti-rubisco activase antibody to label the chloroplasts of *E. gracilis* cells, and attempted to measure the change in size and morphology of these plastids during different phases of the dark/light cycle. While it is not surprising that a morphological change was observed, it seems unlikely that the use of anti-rubisco activase fluorescence would accurately quantify plastid volume and carbon fixation rate, especially given that the authors state that this particular antibody specifically stains the pyrenoids within the plastids and weakly stains the remainder of the plastid, while also non-specifically staining other cellular structures. It would be helpful for these authors to provide a rationale for using this stain and to cite other literature that has successfully used anti-rubisco activase staining to quantify plastid volume and rates of carbon fixation (photosynthesis). These authors also only presented micrographs of one representative cell for each time point at which they were interested in comparing organelle volume and labelling intensity (Supplementary Fig. 3), which does not seem like a large enough sample size (especially when working with a cultivated organism) to claim that there was/wasn't a change in volume or fluorescence intensity. If the number of measured cells was greater, the authors should indicate this in the text.

* The use of terms in the discussion that had not previously been mentioned (e.g., "mastigont") made it challenging to follow these results/findings from the results to the discussion of this manuscript.

Reviewer #2 (Comments to the Authors (Required)):

Expansion microscopy is a great new technique with which to better visualize certain structural features of cells. The application of this technique to the imaging of individual cells represents an exciting new area of research and the application of this technique to a species of Euglenid is the first of its kind. The demonstration that Euglenid cells can be expanded in such a way that both maintains their integrity and normal morphological appearance while at the same time allows for improved imaging of various cellular features (e.g. cytoskeleton, mitochondrion, chloroplasts) is a notable contribution to the field. As articulated below, perhaps the most significant advantage of expansion microscopy is that it will make it far easier to study dynamic processes that occur over the cell cycle. These include changes in the microtubular cytoskeleton, the mitochondrion under different culture conditions, and the size, shape and distribution of chloroplasts. Specific thoughts on each of these are below:

Microtubular Cytoskeleton: While it is apparent to anyone familiar with the cytoskeleton of Euglenids, the authors should amend their descriptions of the pellicular microtubules that comprise the cytoskeleton to clarify the fact that these are actually groups of four closely appressed microtubules. This is apparent from the TEM section in figure 1E but it should be made clear to the reader in the text. Many of the figures (e.g. 1A, 1B, 2A) show the pellicular microtubules as a what appears to be a single line and a reader who is unfamiliar with euglenid cytoskeleton may misinterpret these as representing individual microtubules. For this reason the numbering (1-4) of the four microtubules that comprise a single pellicular strip should be improved, perhaps by the use of black on white numbering instead of the black only numbers that are used in Fig. 1E. Likewise this reviewer had difficulty seeing the "Ax" in figure 1F.

Another point is that TEM images (1E and 1F) represent a cell that was unexpanded. It is important to make this explicitly clear as it has great bearing on how the expansion factor was determined.

The transition zone of each flagellum (which lacks a central pair and associated structural proteins in Euglenids) can also have a slightly expanded appearance when prepared for conventional TEM. This inflated appearance is amplified in the case of expansion microscopy in which the diameter of the transition zone appears to have a slightly inflated diameter than the rest of the complete axoneme. In this reviewer's opinion this is most likely due to the reduced integrity of the transition zone due to the absence of structural components of the axoneme. The observed enlargement of the transition zone is most likely due to the lack of a central pair of microtubules and the associated structural proteins (i.e. radial spokes, inner sheath complex and perhaps Nexin links between doublets). Without these structural proteins to constrain expansion, it is not surprising that the transition zone appears broader in diameter than that of the axoneme and much larger than what has previously been reported from TEM studies. Some mention of the known ultrastructural architecture of the Euglenid flagellar transition zone would be helpful for those readers who are less familiar with this group of protists. While the authors did attempt to do this "The widths of transition zones but perhaps also the distal parts of the basal bodies and proximal regions of the axoneme in both flagella were visibly inflated." I feel that they did not adequately address the underlying basis for this point.

Perhaps the most significant figure in the paper is the series of images in Figure 2; the assembled Z-stack staining for microtubules. Previous studies that attempted to reconstruct the complex microtubular architecture of the flagellar roots, flagellar pocket, and pellicular cytoskeleton, required the laborious technique of serial reconstruction based on a great many TEM images. These were then visually assembled and typically represented as an artist's illustration. Invariably the effort and time involved to create a single three-dimensional structure precluded doing so for more than one cell. Given the dynamic nature of euglenid metabolism the possibility now exists to create similar three-dimensional views for a large number of cells and to image the dynamic relationship between individual pellicular strips. In addition, as the author's exquisitely demonstrate in figures 2C-E that this can also be done for cells at various stages of cell division, something that would have been well beyond the scope of TEM or even conventional confocal microscopy.

Mitochondrion: The observation that the nature of the single mitochondrion changes based on culture condition is yet another unique contribution of this paper. While it may not have been the author's intent to discover this, their observation opens the possibility of future studies. It would be interesting if the technique of expansion microscopy could be applied to study mitochondrial morphology on cells grown under different oxygen levels and/or *E. gracilis* cells grown axenically in complete darkness.

Chloroplasts: As with the mitochondrion, this study suggests future areas of research in which dynamic changes in chloroplast structure, number, and distribution might be studied under varying culture/cell cycle conditions. As I am sure is the case with the authors, this reviewer was disappointed to see that the normal autofluorescence of the chloroplasts was not retained during the processing of the cells. It would be interesting to know the author's opinion as to why this was the case. Regardless, the authors clearly demonstrate the utility of using expansion microscopy to study these organelles in a large number of cells under a variety of culture and environmental conditions.

Summary: In summary this reviewer finds this paper to be well crafted, carefully done, and significant contribution the study of single celled protists, not merely Euglenids. Thus it should be of broad interest to wide number of researchers and I recommend publication with the minor revisions noted above.

Reviewer #3 (Comments to the Authors (Required)):

This is an important study. To date IF has been impossible in Euglena due to the pellicle hindering access. The authors have surmounted this very well and provide convincing data. My comments are for minor modifications.

At first, we assessed three solvents (PBS, 25mM sucrose buffer and growth media) for fixation, of which the latter resulted in the best-preserved cell shape in all media tested-Cramer-Myers media (CMM), CMM with 0.4% (v/v) ethanol (CMMEt) and Hutner's media (HM)-and was further used for all experiments. Unless stated otherwise, we used the cells grown in CMMEt.

>Need referencing. Strictly these are buffers and not solvents.

For each measured cell, 20 values were taken and the average value was used; each N used for the final value is represented by this average.

>Important to look at individual numbers to ascertain variability?

While best documented I would ask what can you do to look at expansion of less rigid structures beyond the cytoskeleton? Question here is what happens to more flexible things such as ER, Golgi, nucleus etcetera and is this even and in line with cytoskeletal expansion.

Visualisation of the microtubular cytoskeleton/Morphology and development of mitochondria/Morphology and development of chloroplasts

>Can the findings here be compared at all with other protists and with prior work on Euglena?

>Also, there is no attempt to interpret the morphology changes during aging cultures, which detracts somewhat from the data. This would be a lot more interesting if some explanation of what the authors think may be happening were included. I'm a bit unclear here, especially w/ the plastid as the cells are in a 12hr light/dark cycle, so what drives the time dependant changes here?

>Sources of antibodies and validation by Western?

while the expansion factor of the gel (4.5x) was almost identical to the reported by the previous study (4.7x).

>Specific reference needed here

however, the consistent results in the case of chloroplasts indicate that the methodology is robust,

>Or preserves selected structures?

Dear Editor,

Thank you and the reviewers for your positive evaluation of our manuscript and for your constructive comments. We have carefully considered your suggestions and incorporated them into the revised version of the manuscript. We also included new references, the characterization of antibodies by co-IP and WB (Supplementary file 2) plus methodology and tomography of the mitochondria (Supplementary video 2) as requested by reviewers. Additionally, we conducted further experiments using two newly available antibodies that specifically label the thylakoid membranes of chloroplasts. One of these antibodies, which targets the C core subunit of photosystem I, produced a specific signal. Although we have decided not to include this data in the manuscript due to unsuccessful preparations under certain conditions, it has helped us confirm several of our conclusions regarding plastid morphology and size. Finally, we have intensively edited the manuscript to improve the language and style.

Please find our detailed responses to all comments in bold text below. We hope that the revised manuscript will be deemed suitable for publication in the Life Science Alliance.

Sincerely,

Anežka Konupková, Priscila Peña-Díaz and Vladimír Hampel

Reviewer #1

The study successfully applied expansion microscopy as a technique to visualize immunofluorescently labelled structures in *E. gracilis* at a level of detail and resolution that is comparable to transmission electron microscopy. The authors modified an existing expansion microscopy protocol to better preserve cell shape and ultrastructure. They labelled the cytoskeleton, mitochondria and chloroplasts of cultivated *E. gracilis* cells and described traits structures, such as the flagellar root system, that were consistent with previous electron microscopy studies of *E. gracilis*. This paper illustrates that it is feasible to use expansion microscopy as a method to visualize different cytoskeletal elements in cultivated euglenids, possibly in substitution for TEM, which is exciting.

Thank you for your positive words.

However, the manuscript was not thoroughly edited prior to submission; it lacked clarity throughout and contained several spelling errors and grammatical mistakes (e.g., sentence structure, paragraph structure). It might be helpful for the authors to work with a professional editing service to revise the text and incorporate adequate scientific background for the results and discussion.

We have carefully revised the manuscript to enhance grammar and style with the assistance of AI text editors. We hope that the new version reflects this improvement and is suitable for publishing.

As described below, I am suspicious of some of the claims made by the authors, in part due to an obvious lack of relevant literature cited and a rationale for the specific techniques/experiments performed (e.g., comparisons of culture media, light regimes etc.). While these authors present beautiful and confirmatory micrographs of the cytoskeleton and organelles of *E. gracilis*, the manuscript is limited in terms of novel results.

My specific comments/suggestions are provided below:

Introduction

The introduction of this manuscript is missing a significant amount of relevant information that would greatly improve the clarity of the paper. For example, the authors highlight that using expansion microscopy on *E. gracilis*, an organism that has been extremely well-characterized, could be useful, but they do not indicate any specific findings/ultrastructural traits/questions etc. that can be elucidated using expansion microscopy. From the introduction, the aim of this paper is unclear. It appears that these authors are trying to present a new protocol for expansion microscopy that can be successfully applied to visualizing the cytoskeleton, mitochondria and plastids of *E. gracilis*. There is no mention of testing various media/fixation buffers, or observing changes in organelle morphology during different phases of a light/dark cycle. I suggest that these authors flesh out the introduction with more background information to provide the necessary context for the rest of the paper and to highlight the aims of this study. For example, why might light/dark cycles influence mitochondrial ultrastructure? Why test this? This type of information should be made clear in the introduction of this manuscript.

A brief commentary on the metabolic flexibility of *Euglena*, the tomography of its organelles, and the potential benefits of expansion microscopy have been added to the end of the introduction. We have also clearly specified the aims of our study:

“The main aim of this study was to validate expansion microscopy as a method for immunolabelling internal structures in *Euglena* and explore its usefulness for monitoring changes in the size and shape of chloroplasts and mitochondria.”

The literature cited in the introduction of the paper is limited, where several important statements are missing any citations, and some statements are missing key citations

for relevant papers. For example:

"This technique has proved useful in visualising the spatial organisation of the cytoskeleton, invasion apparatus, and the localisation of the translocons on the membrane of the hydrogenosomes (mitochondrion-related organelles)."

Including citations for each of these three examples and providing information on what organisms were used for each example (i.e., the invasion apparatus of which organism, hydrogenosomes of which organism).

The citations were provided after the preceding sentence, which we acknowledge was confusing. To clarify this section, we have rewritten it and included a citation for a new manuscript published in BioRxiv during the review process. The revised text reads as follows:

"It has been successfully applied to mammalian cells and tissues (Hümpfer et al., 2024; Chen et al., 2015; Ku et al., 2016) and to many single-cell eukaryotes across the eukaryotic tree including genera Giardia, Plasmodium, Trypanosoma, Leishmania, Chlamydomonas, Trichomonas, Eutreptiella and various dinoflagellates. In these microbial eukaryotes this technique has proved useful in revealing the spatial organisation of their cytoskeleton, invasion apparatus, and the localisation of the translocons on the membrane of the hydrogenosomes, organelles related to mitochondria (Halpern et al., 2017; Gambarotto et al., 2019; Bertiaux et al., 2021; Gorilak et al., 2021; Makki et al., 2024; Mikus et al., 2024). ,,"

"The presence of this surface barrier, however, complicates the staining of internal structures with antibodies, making fluorescence microscopy of little use for this organism."

Provide citations for this statement. I would also suggest providing some information on how expansion microscopy mitigates this problem.

Currently, there are no studies that investigate the reasons for the limited success of immunofluorescence staining of internal structures in *E. gracilis*. The lack of immunofluorescence images of the internal structures of *Euglena* in the literature highlights this issue. While there have been a few studies focusing on immunolabeling cytoskeletal proteins (Mermelstein et al., 1998) and flagellar proteins (Nasir et al., 2018), these studies primarily targeted either flagella or the pellicle. We dedicated considerable time to optimizing the classical immunofluorescence microscopy conditions for *E. gracilis*, experimenting with various fixative solutions, pre-fixing techniques, and buffers for denaturation and staining using many different antibodies.

Unfortunately, these efforts resulted in little success. We have rewritten the part of the manuscript to better explain the problem.

“It is likely that the presence of this surface barrier prevents the staining of internal structures, apart from the pellicle and flagellum (Mermelstein et al., 1998; Nasir et al., 2018), with antibodies. This is evident from the absence of immunofluorescence studies in this organism.”

"The organisation of euglenid cytoskeleton..." and "The pellicle is a complex..."

Both of these sentences are missing several important citations that specifically address the unity, diversity and evolutionary history of traits associated with the pellicle.

We apologize for missing some important references; the older literature on this topic is extensive. We have updated the list of references currently cited regarding the organization of the *Euglena* cytoskeleton, which includes: Piccinni and Mammi (1978), Surek and Melkonian (1986), Farmer and Triemer (1988), Shin et al. (2002), Willey and Wibel (1985). Additionally, for the pellicle structure, we have included the following references: Cavalier-Smith (2017), Leander et al. (2007), Esson and Leander (2006), Sommer (1965), Bricheux and Brugerolle (1987), Dubreuil and Bouck (1988). If the reviewer feels that we have still overlooked an important reference, please let us know, and we would be happy to include it.

Results & Methods:

** The first two paragraphs of the results section read as a methods section.*

We agree, however, as this is the first application of expansion microscopy on *E. gracilis*, we decided to include more details on the optimization part in the result section. We believe it is valuable to provide this context. Also, the determination of the expansion factor is an important result, which also allows the comparison with the previously reported studies on other organisms.

** The figures presented in this manuscript are of high quality and highlight the cytoskeletal and organellar complexity of *E. gracilis*. Many of the labels used in these figures are not cited in the text, and as a result, it was challenging to follow the text at times. Consider including a reference to the labels in the figures consistently throughout the text of the Results section.*

Thank you for pointing this out. The references were added to the text in the result section describing the organisation of the cytoskeleton. We hope the text is now easier to follow.

** The authors attempted to quantify ultrastructural differences in mitochondria and*

plastids as well as changes in carbon fixation rates by plastids, both in response to light limitation by fluorescently labelling these structures and imaging them via expansion microscopy.

It is strange that the authors did not observe any autofluorescence signal from the plastids of the cells given that *E. gracilis* cells are well known to autofluoresce, and no clearing step was indicated in the methods (as this would be necessary to clear the fixed cells of autofluorescence). Was a negative control used to determine whether or not there was autofluorescence present? Because chlorophyll autofluorescence would be observed using the excitation wavelengths that these authors used (488 nm), a negative control would be necessary to determine if there is/isn't autofluorescence. Including negative controls when performing fluorescence microscopy experiments is a standard for these types of experiments to substantiate the claim that autofluorescence was not present in the tissue/cells imaged.

Negative controls were performed for all the experiments but we were not reporting them. We understand that this may be confusing so we now added negative controls to Supplementary Figure 2 for the integrity of the results.

The autofluorescence was, indeed, not observed. No extra clearing step was used for that, the procedure was done exactly as is specified in the Method section. Apparently, the chlorophyll autofluorescence does not survive the treatment during the expansion microscopy protocol. It remains unclear why this is so, but it is consistent with the study of Mikus et al. (2024) using expansion microscopy, where they also do not observe any autofluorescence.

* The authors specified that they used AlexaFluor 488 dye for each secondary antibody, meaning that the cytoskeleton, mitochondria and plastids of the prepared cells would all be labelled with the same fluorophore and would fluoresce under the same excitation spectra. If this is the case, describe how it was possible to obtain micrographs of the plastids, mitochondria and microtubular structures separately for the same cells - as it seems as though all three structures would fluoresce at once. Could this be an issue in how these methods were presented?

Thank you for noticing this, it was a typo. We used AlexaFluor 488 dyes for the mitochondria, plastids and some of the cytoskeletal samples, however, Alexa Fluor™ 594, goat anti-guinea pig IgG was used for the α Tub visualisation of the cytoskeleton when combined with the antibody against mitochondria or chloroplasts markers. This mistake was corrected.

* The authors test several different media for fixation to select one that results in optimal ultrastructural preservation of cells; however, they do not show any micrographs of these different tests as part of the results. If the authors wish to conclude that one type of media is better at preserving cell morphology than others, it

would be good to include micrographs that show the differences in cell preservation using the different types of media as supplementary material.

As described at the beginning of the Results, we tested fixing the cells in two buffers, PBS (used in Gorilak et al., 2021) and sucrose buffer (Heiss et al., 2024 - doi.org/10.1101/2024.03.29.587336), however, this resulted in either not expanded or broken cells. The best results were achieved when the cells were fixed in the same media, in which they grew, being it CMM, CMMEt or HM. These cells expanded, had intact pellicles and most often even with flagella still attached to them. We do not report the images from the suboptimal fixes, because we do not want to extend the picture documentation and we consider it sufficient to mention this in the text. If the reviewer insists we can prepare a supplementary figure with these suboptimal experiments.

* The language used in the text is not always consistent with the literature. For example, using terms such as "turn/turning" or "dorso-anteriorly" when describing the flagellar root system is not consistent with the language used in other ultrastructural studies of euglenids. *The authors also interchange the terms "immunolabeled" with "immunodecoration" and "immunostain". It would be clearer to maintain consistency with the terms used throughout the text.*

The "immunodecoration" was replaced by "immunostain". We also avoided using turn/turning but we kept the term dorso-anteriorly, because we find it a standard description of the direction in the cell and do not know, how to replace it. The text now reads:

"The VR initially travels in the ventral-left direction and then loops around the bottom of the flagellar pocket directing right-anteriorly and flanking the flagellar pocket in the region free of pellicular microtubules. The distal part of the VR detaches from the flagellar pocket and stretches dorso-anteriorly towards the pellicle (Figure 2A, B section 8). The IR originates in the region between the two basal bodies, travels leftwards and then continuously changes its course in a right-anterior direction around the flagellar pocket. A sheet of ventral pellicular microtubules (VPM) originates in the proximity of the distal part of the IR and supports the membrane at the ventro-anterior side of the pocket (Figure 2A, B sections 9-10). The dorsal root (DR) originates at the dorsal side of the basal body 2 (Figure 2 B section 1), extends leftwards and for some distance travels closely along the IR (Figure 2A, B sections 3-7)."

* *The text describing Figure 2C is limited to only one sentence. Consider describing these results in more detail.*

A more detailed description of Figures 2 C-E was included:

"Figures 2C-E display mitotic cells in various stages. In these cells, the rows of pellicular microtubules are doubled to be distributed into daughter cells upon cytokinesis. In Figure 2C, the mitotic spindle (MS) is formed and the flagella are multiplied to four, all within a single flagellar pocket, of which only one, presumably the old anterior flagellum, reaches out of the canal. In Figures 2D and E, the flagellar pocket divides but remains connected in the canal region. The young anterior flagellum continuously grows through the canal to reach the outside of the cell. The mitotic spindle divides the chromosomes, which are not stained."

Discussion

* The discussion of this manuscript either does not mention several findings that were highlighted in the results or only superficially attempts to place these results into the context of existing literature. For example, the localization of the mitochondria to the cell surface (in cells that are grown, I assume, aerobically) is unusual and, as far as I know, inconsistent with the previous EM characterizations of aerobic *E. gracilis*.

All *E. gracilis* cells in our experiments were grown aerobically. Thank you for pointing out the observation of the position of mitochondria, which we did not discuss. Pellegrini (1980) reported that in the cells grown photoautotrophically in CMM, the mitochondrion appears as "...a reticulum of threads which are delicate, flattened (0-4 x 0-6 / μ m in thickness), variously branched, extended throughout the cell...". On the other hand, in photoheterotrophically grown cells on CMM with acetate, mitochondrion ,, ...become thicker and more united, flatten tangentially to the cell surface and constitute a dense parietal network with narrow meshes (compare Figs. 14 and 15), This network spreads deeply within the cell to enclose chloroplasts, nucleus and reservoir (Figs. 16, 17)". Our observations confirm that mitochondrion in photoautotrophic cells (CMM) is a more delicate reticulum homogeneously distributed in the volume of the cell. In the heterotrophically grown cells (HM and CMMEt) the mitochondrion indeed becomes thicker and parietal in a close relation with the chloroplasts, which are mostly under the mitochondria. There are some parts of the reticulum that penetrate further into the cell in all media tested. However, the main localization of the mitochondria is above other organelles. The difference from Pellegrini (1980) may be caused by the usage of different carbon sources (ethanol vs acetate) or by the fact that Pellegrini synchronised the culture using cold/warm cycles while our culture was grown in a stable temperature of 25 °C was used. To support our claim we inspected more cells and prepared

illustrative animation (Supplementary Video 2). Furthermore, we also have TEM micrographs confirming the mitochondria localization mostly under the cell surface in the case of CMMEt in our study (not shown).

Discussion on this topic was added to the manuscript.

* These authors used an anti-rubisco activase antibody to label the chloroplasts of *E. gracilis* cells, and attempted to measure the change in size and morphology of these plastids during different phases of the dark/light cycle. While it is not surprising that a morphological change was observed, it seems unlikely that the use of anti-rubisco activase fluorescence would accurately quantify plastid volume and carbon fixation rate, especially given that the authors state that this particular antibody specifically stains the pyrenoids within the plastids and weakly stains the remainder of the plastid, while also non-specifically staining other cellular structures. It would be helpful for these authors to provide a rationale for using this stain and to cite other literature that has successfully used anti-rubisco activase staining to quantify plastid volume and rates of carbon fixation (photosynthesis). These authors also only presented micrographs of one representative cell for each time point at which they were interested in comparing organelle volume and labelling intensity (Supplementary Fig. 3), which does not seem like a large enough sample size (especially when working with a cultivated organism) to claim that there was/wasn't a change in volume or fluorescence intensity. If the number of measured cells was greater, the authors should indicate this in the text.

The anti-RCA antibody is currently the best marker for chloroplasts of *E. gracilis* available to us and we believe it is suitable for the testing of this methodology. It labels the whole volume of chloroplasts and even more intensively stains pyrenoids. The visualization of intensity and morphology pyrenoids is an added value demonstrating the power of expansion microscopy to label proteins inside of the membranous organelles in *E. gracilis*.

The methodology of the volume measurement was firstly optimised on the CMMEt samples with various settings of the intensity thresholds using Nis Elements Software to make sure that the whole volume of the organelle is considered. These pilot tests were done on 10 different cells. In the figure, only one representative cell for each medium is selected, but for the volume measurement, at least 6 cells per media were used (Supplementary file 1, Table 5). The threshold was used the same for all the measured cells except for the CMM cells in which the signal was overall stronger. Cells with a non-specific signal outside chloroplasts were excluded from the measurements.

The measurement of the relative chloroplast volumes using the anti-RCA antibody (or any other antibody) is reported by us here for the first time. To our

knowledge, the only measurements of chloroplasts in *E. gracilis* cells were done by Pellegrini (1980a,b) using electron micrographs. We believe that our approach, although it may be less precise, is robust and suitable for routine measurements, which opens a path for future physiological studies linking metabolic and morphological features.

Finally, we also tested a newly acquired antibody against the C core subunit of the photosystem I (AS10 939, Agrisera) that labelled thylakoid membranes and measured them with the same methodology for 5 cells from the CMMEt media with the same conditions used for the cells in this study and the percentage of the relative chloroplasts volume in the cells was comparable, 5.12 – 7.92 % (5.33-8.72 with anti-RCA) validating the usage of anti-RCA. We did not include results from this antibody in the manuscript as we do not have good preparations under all conditions.

* The use of terms in the discussion that had not previously been mentioned (e.g., "mastigont") made it challenging to follow these results/findings from the results to the discussion of this manuscript.

The term mastigont was removed.

Reviewer #2 (Comments to the Authors (Required)):

Expansion microscopy is a great new technique with which to better visualize certain structural features of cells. The application of this technique to the imaging of individual cells represents an exciting new area of research and the application of this technique to a species of Euglenid is the first of its kind. The demonstration that Euglenid cells can be expanded in such a way that both maintains their integrity and normal morphological appearance while at the same time allows for improved imaging of various cellular features (e.g. cytoskeleton, mitochondrion, chloroplasts) is a notable contribution to the field. As articulated below, perhaps the most significant advantage of expansion microscopy is that it will make it far easier to study dynamic processes that occur over the cell cycle. These include changes in the microtubular cytoskeleton, the mitochondrion under different culture conditions, and the size, shape and distribution of chloroplasts. Specific thoughts on each of these are below:

Microtubular Cytoskeleton: *While it is apparent to anyone familiar with the cytoskeleton of Euglenids, the authors should amend their descriptions of the pellicular microtubules that comprise the cytoskeleton to clarify the fact that these are actually groups of four closely appressed microtubules. This is apparent from the TEM section in figure 1E but it should be made clear to the reader in the text. Many of the figures (e.g. 1A, 1B, 2A) show the pellicular microtubules as a what appears to*

be a single line and a reader who is unfamiliar with euglenid cytoskeleton may misinterpret these as representing individual microtubules.

Thank you for pointing out this unclarity. We have now clarified it in the text.

„By the single row of pellicular microtubules we mean a group of four microtubules localised at the joint of neighbouring pellicular strips (Figure 1E; Cavalier-Smith, 2017; Bricheux and Brugerolle, 1987). These four microtubules appear in expansion microscopy images as a single line and the distance between lines represents the width of pellicle strips.“

For this reason the numbering (1-4) of the four microtubules that comprise a single pellicular strip should be improved, perhaps by the use of black on white numbering instead of the black only numbers that are used in Fig. 1E. Likewise this reviewer had difficulty seeing the "Ax" in figure 1F. Another point is that TEM images (1E and 1F) represent a cell that was unexpanded. It is important to make this explicitly clear as it has great bearing on how the expansion factor was determined.

Figures 1E-F were corrected according to your suggestions. We have increased the contrast and modified the labels so that they are more visible. We hope the quality is now satisfactory. We have also stressed in the text that the electron microscopy images are taken from unexpanded cells.

“These values were compared with the dimensions measured on electron micrographs of unexpanded, chemically fixed cells (Figure 1E, F). “

The transition zone of each flagellum (which lacks a central pair and associated structural proteins in Euglenids) can also have a slightly expanded appearance when prepared for conventional TEM. This inflated appearance is amplified in the case of expansion microscopy in which the diameter of the transition zone appears to have a slightly inflated diameter than the rest of the complete axoneme. In this reviewer's opinion this is most likely due to the reduced integrity of the transition zone due to the absence of structural components of the axoneme. The observed enlargement of the transition zone is most likely due to the lack of a central pair of microtubules and the associated structural proteins (i.e. radial spokes, inner sheath complex and perhaps Nexin links between doublets). Without these structural proteins to constrain expansion, it is not surprising that the transition zone appears broader in diameter than that of the axoneme and much larger than what has previously been reported from TEM studies. Some mention of the known ultrastructural architecture of the Euglenid flagellar transition zone would be helpful for those readers who are less familiar with this group of protists. While the authors did attempt to do this "The widths of transition zones but perhaps also the distal parts of the basal bodies and proximal regions of the axoneme in both flagella were visibly inflated." I feel that they did not adequately address the underlying basis for this point.

We would like to thank the reviewer for this insight. Honestly, we were not aware of this connection but it makes a lot of sense. We now discuss this point

and include references to support it. We took the liberty to use some of the expressions used by the reviewer

“It is known that the transition zones of Euglenida are wider, hollow and relatively long (Piccinni and Mammi, 1978; Farmer and Triemer, 1988; Simpson, 1997; Kivic and Walne, 1984). The absence of internal structural components in their interior likely contributes to a decrease in structural integrity, resulting in a disproportionate broadening during expansion.”

Perhaps the most significant figure in the paper is the series of images in Figure 2; the assembled Z-stack staining for microtubules. Previous studies that attempted to reconstruct the complex microtubular architecture of the flagellar roots, flagellar pocket, and pellicular cytoskeleton, required the laborious technique of serial reconstruction based on a great many TEM images. These were then visually assembled and typically represented as an artist's illustration. Invariably the effort and time involved to create a single three-dimensional structure precluded doing so for more than one cell. Given the dynamic nature of euglenid metaboly the possibility now exists to create similar three-dimensional views for a large number of cells and to image the dynamic relationship between individual pellicular strips. In addition, as the author's exquisitely demonstrate in figures 2C-E that this can also be done for cells at various stages of cell division, something that would have been well beyond the scope of TEM or even conventional confocal microscopy.

Thank you for your positive opinion.

Mitochondrion: The observation that the nature of the single mitochondrion changes based on culture condition is yet another unique contribution of this paper. While it may not have been the author's intent to discover this, their observation opens the possibility of future studies. It would be interesting if the technique of expansion microscopy could be applied to study mitochondrial morphology on cells grown under different oxygen levels and/or *E. gracilis* cells grown axenically in complete darkness.

Chloroplasts: As with the mitochondrion, this study suggests future areas of research in which dynamic changes in chloroplast structure, number, and distribution might be studied under varying culture/cell cycle conditions. As I am sure is the case with the authors, this reviewer was disappointed to see that the normal autofluorescence of the chloroplasts was not retained during the processing of the cells. It would be interesting to know the author's opinion as to why this was the case. Regardless, the authors clearly demonstrate the utility of using expansion microscopy to study these organelles in a large number of cells under a variety of culture and environmental conditions.

We agree with the reviewer that this technique opens the possibility to study morphological responses to various physiological conditions. Although it is appealing, the highly variable metabolism of *E. gracilis* makes it unfeasible to test all the possible growth conditions as a part of this paper, in which we

focused on the method itself. We tried at least to demonstrate the sensitivity of the expansion microscopy to detect differences between cells grown on commonly used media.

The missing autofluorescence was one of the motivations to test various combination of fixation time, fixatives, buffers and media. Unfortunately, we did not find a condition under which autofluorescence would be preserved. The reason for the absence of autofluorescence remains unclear but it might be caused by the destruction of the chlorophyll molecules by the harsh conditions during the denaturation with SDS and boiling for one hour at 95°C.

Reviewer #3 (Comments to the Authors (Required)):

This is an important study. To date IF has been impossible in Euglena due to the pellicle hindering access. The authors have surmounted this very well and provide convincing data. My comments are for minor modifications.

Thank you for all your constructive comments. We modified the manuscript accordingly to them.

“Cramer-Myers media (CMM), CMM with 0.4% (v/v) ethanol (CMMEt) and Hutner's media (HM)-and was further used for all experiments. Unless stated otherwise, we used the cells grown in CMMEt.

>Need referencing. Strictly these are buffers and not solvents”.

The referencing is needed indeed. They are specified in the Material and Methods section. To avoid confusion we add them also to the results as requested. We replaced “solvents” by “buffers”.

For each measured cell, 20 values were taken and the average value was used; each N used for the final value is represented by this average.

>Important to look at individual numbers to ascertain variability?

We agree and it was our intention from the very beginning to provide as much raw data as possible. Standard deviations (SD) for measurements of every individual cell are therefore provided in Supplementary file 1 (columns C and H).

While best documented I would ask what can you do to look at expansion of less rigid structures beyond the cytoskeleton? Question here is what happens to more flexible things such as ER, Golgi, nucleus etcetera and is this even and in line with cytoskeletal expansion.

Given the protocol, during which the sample is fixed and the biomolecules are anchored in the hydrogel prior to denaturation, the expansion of all structures should theoretically be roughly proportionate. However, this is not always the case as demonstrated by the increased expansion of transition zones of flagella. Unfortunately, we can visualise only structures for which we have labels and we do not possess any specific antibody for the ER or Golgi in our laboratory for *E. gracilis*. On the other hand, plastids and particularly mitochondria, which were the focus of this study, are also delicate, non-rigid structures. Their expansion seemed by eye roughly proportionate to the cytoskeleton but we do not know how to determine this objectively on such complicated objects. In addition, the visualisation of the pyrenoid, which is localised within plastids and not separated by any membrane, testifies that non-membraneous bodies also expand roughly proportionately. Again, an objective assessment of the expansion factor and the evenness of expansion of pyrenoid was not done. We did not include the nucleus staining in our study, but there is a new preprint showing a cytoskeleton and the nucleus expansion in various protists, including *Eutreptiella* which also possesses a rigid pellicle (Mikus et al., 2024). We believe that the nuclei of *E. gracilis* would be preserved and expanded evenly.

Visualisation of the microtubular cytoskeleton/Morphology and development of mitochondria/Morphology and development of chloroplasts

>Can the findings here be compared at all with other protists and with prior work on Euglena?

>Also, there is no attempt to interpret the morphology changes during aging cultures, which detracts somewhat from the data. This would be a lot more interesting if some explanation of what the authors think may be happening were included. I'm a bit unclear here, especially w/ the plastid as the cells are in a 12hr light/dark cycle, so what drives the time dependant changes here?

The expansion microscopy is quite novel in the field of protistology and therefore there are not many studies to compare with. The closest models are Kinetoplastids (Gorilak et al, 2021), which we referred to. In October 2024 a preprint from Mikus et al. (doi: <https://doi.org/10.1101/2024.10.18.618984>) was released, showing a cytoskeletal diversity among the major protist groups by expansion microscopy, including *Eutreptiella gymnastica* and *Eutreptiella braarudii* from the group Euglenida. The displayed pellicle and flagellar pockets are superficially similar to Euglena but they did not attempt to show and describe any details, because the study is taxonomically broad and euglenids are not the main focus of it. Mitochondria and chloroplasts were visualized neither in Gorilak nor in Mikus papers. Our findings can be compared with the 3D reconstruction from the electron micrographs done by Pellegrini, 1980 and this was discussed in our manuscript.

We agree that linking the morphological changes of organelles with the changes in cell metabolism would be interesting. The morphology changes most likely are connected to their metabolic activity, which is affected by the light cycle, the composition of the media, the growth phase, the age of the culture and other factors. These aspects were already proven to have an effect on the cells (Gross and Villaire, 1960 - doi.org/10.2307/3224080; Wang *et al.*, 2018 - doi: 10.1371/journal.pone.0195329) and different shapes of the chloroplasts were observed in the electron micrographs during the dark phase and 1-2 hours after the beginning of the light phase (Cook *et al.*, 1976 - doi: 10.1111/j.1550-7408.1976.tb03790.x). Unfortunately, experiments which could start disentangling these relationships would require transcriptome/proteome data from the time points, and biochemical measurements and this way exceeded the scope of this manuscript. For exactly this reason we stayed brief in any discussion regarding this matter.

>Sources of antibodies and validation by Western?

Antibodies used in the work are specified in the Material and Methods section, two of them are commercial (anti- α -tubulin and anti-rubisco activase) and all of them are either used previously (anti- α -tubulin e.g. in Heiss *et al.*, 2024 - doi.org/10.1101/2024.03.29.587336) or verified by Western blot analysis and co-immunoprecipitation followed by mass spectrometry (anti- β -ATP synthase and α -Ribulose-1,3-BisPhosphate activase). Co-immunoprecipitation showed that anti- α RCA recognises mainly RCA signal, marginally chloroplast photosystem II protein M precursor and light-harvesting chlorophyll a /b binding protein of PSII. Anti- β -ATPase recognises mainly the D-chain of mitochondrial ATP synthase beta subunit and chloroplastic ATP synthase CF1 beta subunit and marginally various chloroplast or mitochondrial proteins, and tubulins. There are various unspecific signals in western blot analysis of anti- α RCA, however, there is easily distinguishable the strongest signal of expected size approximately 50 kDa. Anti- β -ATPase shows a strong signal of solely two bands above and under 55 kDa corresponding with the mass spectrometry measured mitochondrial ATPase (56.5 kDa) and chloroplast ATPase (52.1 kDa). We included these data in Supplementary File 2.

while the expansion factor of the gel (4.5x) was almost identical to the reported by the previous study (4.7x).

>Specific reference needed here

The text refers to the same study used in the first part of the sentence to avoid confusion the reference was moved to the end of the sentence.

however, the consistent results in the case of chloroplasts indicate that the methodology is robust,

>Or preserves selected structures?

The reviewer is correct that the two structures mitochondria vs. chloroplast may theoretically differ in their sensitivity to artifacts in expansion microscopy. We have reformulated this part of the discussion to admit also this option.

“We cannot rule out the possibility that the variation in volumes of mitochondria may have been due to a methodological artifact. Yet, consistent volumes of chloroplasts suggests that either the expansion factor of mitochondria varies markedly more than that of chloroplast, or that the cells in the population exhibit considerable variability in mitochondrial volumes. We favour the latter, considering that the population was asynchronous and included cells at different stages of their life cycle.”

January 27, 2025

RE: Life Science Alliance Manuscript #LSA-2024-03110-TR

Dr. Vladimír Hampl
Charles University
Parasitology Department
Albertov 6
Prague 2 12800
Czech Republic

Dear Dr. Hampl,

Thank you for submitting your revised manuscript entitled "Visualization of *Euglena gracilis* Organelles and Cytoskeleton Using Expansion Microscopy". We would be happy to publish your paper in Life Science Alliance pending final revisions necessary to meet our formatting guidelines.

- please be sure that the authorship listing and order is correct
- label your supplementary tables as table S1-S7...both when labeling them in the system and the manuscript text, their legends, and callouts
- please add the Twitter/X and Bluesky handles of your host institute/organization as well as your own or/and one of the authors in our system
- please mark the corresponding author on the manuscript file
- there is a callout for Figure S4, and this figure has not been uploaded -- please correct
- please add callouts for Figures S2C and S3 to your main manuscript text

A. FINAL FILES:

B. MANUSCRIPT ORGANIZATION AND FORMATTING:

Thank you for your attention to these final processing requirements. Please revise and format the manuscript and upload materials within 5 days.

Sincerely,

January 29, 2025

RE: Life Science Alliance Manuscript #LSA-2024-03110-TRR

Dr. Vladimír Hampl
Charles University
Parasitology
Průmyslová 595
Vestec 25250
Czech Republic

Dear Dr. Hampl,

Thank you for submitting your Methods entitled "Visualization of *Euglena gracilis* Organelles and Cytoskeleton Using Expansion Microscopy". It is a pleasure to let you know that your manuscript is now accepted for publication in Life Science Alliance. Congratulations on this interesting work.

DISTRIBUTION OF MATERIALS:

Again, congratulations on a very nice paper. I hope you found the review process to be constructive and are pleased with how the manuscript was handled editorially. We look forward to future exciting submissions from your lab.

Sincerely,
